# Review for Rare-Earth-Modified Perovskite Materials and Optoelectronic Applications

**DOI:** 10.3390/nano12101773

**Published:** 2022-05-23

**Authors:** Bobo Li, Feng Tian, Xiangqian Cui, Boyuan Xiang, Hongbin Zhao, Haixi Zhang, Dengkui Wang, Jinhua Li, Xiaohua Wang, Xuan Fang, Mingxia Qiu, Dongbo Wang

**Affiliations:** 1College of New Materials and New Energies, Shenzhen Technology University, Shenzhen 518118, China; libobo@sztu.edu.cn (B.L.); 2070413004@stumail.sztu.edu.cn (X.C.); xiangboyuan@sztu.edu.cn (B.X.); 2State Key Laboratory of High Power Semiconductor Lasers, School of Physics, Changchun University of Science and Technology, Changchun 130012, China; tianfengcust@163.com (F.T.); wccwss@foxmail.com (D.W.); lijh@cust.edu.cn (J.L.); biewang2001@126.com (X.W.); 3State Key Laboratory of Advanced Materials for Smart Sensing, General Research Institute for Nonferrous Metals, Beijing 100088, China; zhaohongbin@grinm.com; 4School of Science and Engineering, The Chinese University of Hong Kong, Shenzhen 518172, China; zhanghaixi@cuhk.edu.cn; 5Department of Opto-Electronic Information Science, School of Materials Science and Engineering, Harbin Institute of Technology, Harbin 150001, China; wangdongbo@hit.edu.cn

**Keywords:** metal halide perovskite, rare-earth metal, solar cell, light-emitting diode, photodetector, luminescent solar concentrators

## Abstract

In recent years, rare-earth metals with triply oxidized state, lanthanide ions (Ln^3+^), have been demonstrated as dopants, which can efficiently improve the optical and electronic properties of metal halide perovskite materials. On the one hand, doping Ln^3+^ ions can convert near-infrared/ultraviolet light into visible light through the process of up-/down-conversion and then the absorption efficiency of solar spectrum by perovskite solar cells can be significantly increased, leading to high device power conversion efficiency. On the other hand, multi-color light emissions and white light emissions originated from perovskite nanocrystals can be realized via inserting Ln^3+^ ions into the perovskite crystal lattice, which functioned as quantum cutting. In addition, doping or co-doping Ln^3+^ ions in perovskite films or devices can effectively facilitate perovskite film growth, tailor the energy band alignment and passivate the defect states, resulting in improved charge carrier transport efficiency or reduced nonradiative recombination. Finally, Ln^3+^ ions have also been used in the fields of photodetectors and luminescent solar concentrators. These indicate the huge potential of rare-earth metals in improving the perovskite optoelectronic device performances.

## 1. Introduction

In the past 12 years, metal halide perovskite materials with the chemical formula of ABX_3_, where A represents monovalent cation (such as CH_3_NH^3+^, HNCHNH^2+^, Cs^+^), B is divalent metal cation (such as Pb^2+^, Sn^2+^), and X represents halogen anion (such as I, Br and Cl), have triggered an enormous research wave [1,2,3]. Up to now, perovskite materials have been widely used in the field of photoelectronic devices, including solar cells, light-emitting diodes (LEDs), lasers, photodetectors, and so on, which is attributed to the excellent photo-electronic properties of broad absorption spectrum, tunable energy band gap, long charge carrier lifetime and length, low recombination loss and cost-effective preparation technology [4,5,6,7]. Both the power conversion efficiency (PCE) of perovskite solar cells (PSCs) and the external quantum efficiency of perovskite LEDs have exceeded 25% through the efforts of many researchers, which indicates the tremendous potential in the future commercial applications for the perovskite optoelectronic devices.

There have been many reports about improving the performance of perovskite devices. For example, in the field of perovskite solar cells, various methods have been demonstrated, including tuning the perovskite component proportion, controlling the film quality and crystalline size of perovskite active layer, optimizing the device structure, introducing plasmonic nano-converters, and so on [8,9,10,11]. Although these methods have boosted the PCE of PSCs, upper threshold value may be hampered for further development. The Shockley–Queisser limit is the theoretical limit of energy conversion for single-junction solar cells. As perovskite materials have an intrinsic band gap, there are also limits to the efficiency of perovskite solar cells. In order to break Shockley–Queisser limit, it is an effective method to utilize infrared light outside the intrinsic band gap of perovskite by introducing up-conversion or down-conversion materials into the solar cell. Rare-earth (RE) metals with triply oxidized state (lanthanide ions, Ln^3+^) possess different kinds of energy transitions, which determined that they can emit fluorescence in a wide wavelength range covering from ultraviolet to intermediate infrared regions [12]. Herein, Ln^3+^ ions can be doped into semiconductor materials to act as light active centers, and then they can promote the light absorption and tailor the band gap of the host materials, thus playing the role of up-conversion or down-conversion. In addition, integrating Ln^3+^ into metal halide perovskite materials can be a feasible and effective approach because it minimizes the thermalization losses, through removing the load of mismatch between perovskite absorption spectra and the solar spectrum.

Due to these unique features, rare-earth doped into perovskite crystals and devices for optical and electrical control have been widely reported [13,14,15,16,17,18,19,20,21]. In this article, we review the applications of rare-earth metals in the perovskite photoelectric devices from the following parts: Firstly, Ln^3+^ ions can be doped as up-conversion of down-conversion materials, which can enhance the light absorption efficiency covering from ultraviolet (UV) to near-infrared (NIR) range in the perovskite solar cells. Secondly, Ln^3+^ ions can be used as quantum cutting or band gap tuning dopants to realize high photoluminescence quantum yield (PLQY) and tunable luminescence emission at different wavelength, and then can be efficiently used for white light emission. Thirdly, Ln^3+^ can be used for defect passivation or lead substitution in the perovskite devices. Additionally, Ln^3+^ can also be used in the field of photodetectors and luminescent solar concentrators. At the end of this paper, we give a future research outlook of this field via exploring the effective ideas and experiments which should be demonstrated.

## 2. Modified Strategy of Rare-Earth Metal in Perovskite Devices

There are three main ways of Ln^3+^ ion-modified perovskite photoelectric devices, as shown in Figure 1. Different methods have different hosts of rare earth ions and thus have different mechanisms of modified strategy. Figure 1a shows the introduction of a new conversion layer in the perovskite photoelectric device. By doping rare-earth ions into the host, this layer can play the role of up-conversion or down-conversion of light, thus improving the conversion efficiency of the device. The MFY_4_ (M = Li, Na, K) family is a common host for this layer because of its excellent properties of low phonon energy and high transmittance. Figure 1b shows that rare-earth ions are doped into the electron transport layer or the hole transport layer. After doping, this layer can also play the role of conversion layer. Meanwhile, doping component modulates the transport layer, the flat band edge of the electron transport layer moves toward positive, and the electrons at the conduction band minimum are more easily transferred to the electron transport layer. Figure 1c shows that rare-earth ions are doped into the perovskite lattice. Ln^3+^ ion dopant introduces new energy levels, which not only affects the peak position and the intensity of the original perovskite band edge compound luminescence peak, but also produces a new Ln^3+^ ion luminescence peak in photoluminescence spectrum. Ln^3+^-doped perovskite materials also have a good application prospect in various light-emitting devices due to the unique quantum-cutting mechanism. It should be noted that the luminescence of individual perovskite materials can also be improved by rare-earth doping, not only doped into the active layer of perovskite multilayer devices.

## 3. Applications of Ln^3+^ in Perovskite Solar Cells

As it is well-known, due to the intrinsic band gap of perovskite materials, perovskite- based solar cells are usually unable to utilize light beyond the visible region (the range of UV and NIR light), thus limiting further development of the device efficiency. Therefore, rare-earth metals with characteristics of up- or down-conversion fluorescence were introduced into perovskite to improve their availability of the solar spectrum.

### 3.1. Ln^3+^-Based Up-Conversion Materials

Low-energy near-infrared light absorbed by Ln^3+^-based up-conversion nanoparticles (UCNPs) can be converted to high-energy visible light which can be utilized in perovskite solar cells. For example, Yb^3+^, Er^3+^, Tm^3+^, Ho^3+^-doped or co-doped materials have been reported in up-conversion systems to achieve various up-converted fluorescence in the visible spectrum. The MYF_4_ (M = Li, Na, K, Ru, and Cs) family has the characteristics of low phonon energy, high transmittance, and decrease in the influence of defect states, therefore, it has become an excellent host of Ln^3+^. In 2016, the Yb^3+^ and Er^3+^ co-doped LiYF_4_ transparent single crystal was synthesized and placed closed to the side of FTO in the PSCs by Song’s team, which can induce efficient visible red and green emissions under two ranges of 900–1000 nm and 1500–1600 nm, as illustrated in Figure 2a. The up-conversion quantum efficiency of LiYF4:Yb^3+^/Er^3+^ crystal was 5.72% under 980 nm light excitation (6.2 W cm^−2^) and the PCE of LiYF4:Yb^3+^/Er^3+^/FTO/TiO_2_/MAPbI_3_/HTM/Au device was improved [22]. Having chosen the material, the next question was how to incorporate it into the structure of the device. Lin and co-workers fabricated the CH_3_NH_3_PbI_3_-based PSCs by inserting monodisperse NaYF_4_:Yb/Er up-conversion nanoparticles as the mesoporous electrode. A double hydrophilic PAA-b-PEO diblock copolymer was used as the nanoreactor for UCNPs. The incorporation of NaYF_4_:Yb/Er UCNPs effectively reduced the non-absorption photon loss and harvested the NIR solar photons, followed by the absorption of emitted high-energy photons to generate extra photocurrents. The highest efficiency of NaYF_4_:Yb/Er UCNPs-based cell (FTO/compact-TiO_2_/UCNP/CH_3_NH_3_PbI_3_/Spiro-MeOTAD/Ag) was improved from 16.8% to 18.1% compared with the referenced TiO_2_-based device (FTO/compact-TiO_2_/mesoporous-TiO_2_/CH_3_NH_3_PbI_3_/Spiro-MeOTAD/Ag) [23]. Almost at the same time, Que’s group introduced β-NaYF_4_:Yb^3+^/Tm^3+^@NaYF_4_ core-shell (NYF) nanoparticles into mesoscopic TiO2 scaffold layer which was used as an electron transport layer (ETL), thus, to enhance the NIR light harvest in perovskite solar cells (Figure 2b). As a result, under AM1.5G standard sunlight and 980 nm NIR laser irradiation, the best device based on NYF/TiO_2_ ETL exhibited a PCE of 16.9% after the optimization of the film thickness and NYF/TiO_2_ weight ratio, which was 20% higher than the pristine one [24].

In addition to the core-shell structure, the hexagonal nanoprism structure is also a feasible structure. Jang et al. used hexagonal β-NaYF_4_:Yb^3+^: Er^3+^ nanoprisms as up-converting medias to broaden the absorption spectrum range. The highest PCE of 15.98% was achieved via optimizing the amount of nanoprisms in the mesoporous TiO_2_ layer [13]. Subsequently, up-conversion materials began to be introduced into perovskite layer. Β-NaYF4:Yb/Er up-conversion nanocrystals were introduced into perovskite layer to fabricate planar ITO/ZnO/β-NaYF_4_:Yb/Er-CH_3_NH_3_PbI_3_/Spiro-OMeTAD/Ag PSCs, and the device structure diagram can be seen in Figure 2c. The up-conversion nanocrystals were beneficial for the perovskite film growth, resulting in uniform and pinhole-free surface morphology. In addition, the doped β-NaYF_4_:Yb/Er also enabled NIR absorption. Therefore, the highest device PCE increased from 13.46% to 19.70% for the modified device [25]. In order to further enhance the NIR up-conversion efficiency, Cu_2-x_S was explored to serve as an antenna to sensitize Er_2_O_3_ up-conversion nanoparticles, which exhibited an intense localized surface plasmon resonance absorption band at near-infrared wavelengths. Based on this, mCu_2-x_S@SiO_2_@Er_2_O_3_ nanocomposites were prepared and mixed into mesoporous TiO_2_ ETL. The highest up-conversion luminescence was significantly boosted to 14.3% (inner quantum efficiency) and the excitation spectrum was expanded to the range of 800–1600 nm. Due to the electron transfer from oxygen defects to the conduction band of TiO_2_, the photocurrent of the mCu_2-x_S@SiO_2_@Er_2_O_3_-based device increased, resulting in a champion PCE of 17.8% [26]. In order to prevent the contact between the electron transport layer and the electrode, an insulation layer is often introduced to increase the photovoltage, and the up-conversion material can also be integrated to the insulation layer. Zhao et al. doped Ho^3+^ into NaYbF_4_ to form high-fluorescent UCNPs and then mixed it into the ZrO_2_ scaffold layer (Figure 2d). Under the synergistic effect of NaYF_4_:Ho^3+^ and ZrO_2_, additional photocurrent and photovoltage via up-conversion of NIR light to visible light, declined recombination rate and trap-state density, enhanced charge transfer and realized extraction efficiency, resulting in an overall PCE enhancement of 28.8% in the device of FTO/cp-TiO_2_/mp-TiO_2_/mp-ZrO_2_-NaYF_4_:Ho^3+^/FA_0.4_MA_0.6_PbI_3_/C anode [27]. Sebag and cooperators demonstrated the influence of inserting KY_7_F_22_:Yb^3+^,Er^3+^ UCNPs at the front- or rear-side of perovskite layer (FTO/KY_7_F_22_:Yb^3+^,Er^3+^/FA_0.83_Cs_0.17_Pb(I_0.6_Br_0.4_)_3_/Spiro-MeOTAD/Au and FTO/FA_0.83_Cs_0.17_Pb(I_0.6_Br_0.4_)_3_/KY_7_F_22_:Yb^3+^,Er^3+^/Spiro-MeOTAD/Au) through the light-beam-induced current/fluorescence mapping technique, thus, to quantify the optical and electronic contribution of UCNPs. Figure 2e shows the device structure. The mapping results exhibited decreased green/red ration of up-conversion fluorescence spectra, which was attributed to the increased green fluorescence absorption of perovskite, indicating the optical contribution of UCNPs in the PSCs [28].
Figure 2(**a**) The absorption regions of perovskite solar cells in the AM 1.5G solar spectrum via the up-conversion processes (Reprinted/adapted with permission from Ref. [22]. Copyright 2016 American Chemical Society); (**b**) Cross-sectional SEM image and schematic diagram of FTO/compact-TiO_2_/NaYF_4_:Yb/Er-CH_3_NH_3_PbI_3_/Spiro-MeOTAD/Ag (Reprinted/adapted with permission from Ref. [24]. Copyright 2016 RSC Pub); (**c**) Energy transfer illustration in the device of ITO/ZnO/*β*-NaYF_4_:Yb/Er-CH_3_NH_3_PbI_3_/Spiro-OMeTAD/Ag (Reprinted/adapted with permission from Ref. [25]. Copyright 2017 RSC Pub); (**d**) Diagrams for device structure of FTO/cp-TiO_2_/mp-TiO_2_/mp-ZrO_2_-NaYF_4_:Ho^3+^/FA_0.4_MA_0.6_PbI_3_/C and mechanism of NIR light harvesting and up-conversion processes (Reprinted/adapted with permission from Ref. [24]. Copyright 2018 RSC Pub); (**e**) I Sketch maps for device structure of FTO/FA_0.83_Cs_0.17_Pb(I_0.6_Br_0.4_)_3_/Spiro-MeOTAD/Au with inserting KY_7_F_22_:Yb^3+^, Er^3+^ UCNPs at the front- or rear-side of perovskite layer (Reprinted/adapted with permission from Ref. [28]. Copyright 2018 American Chemical Society).
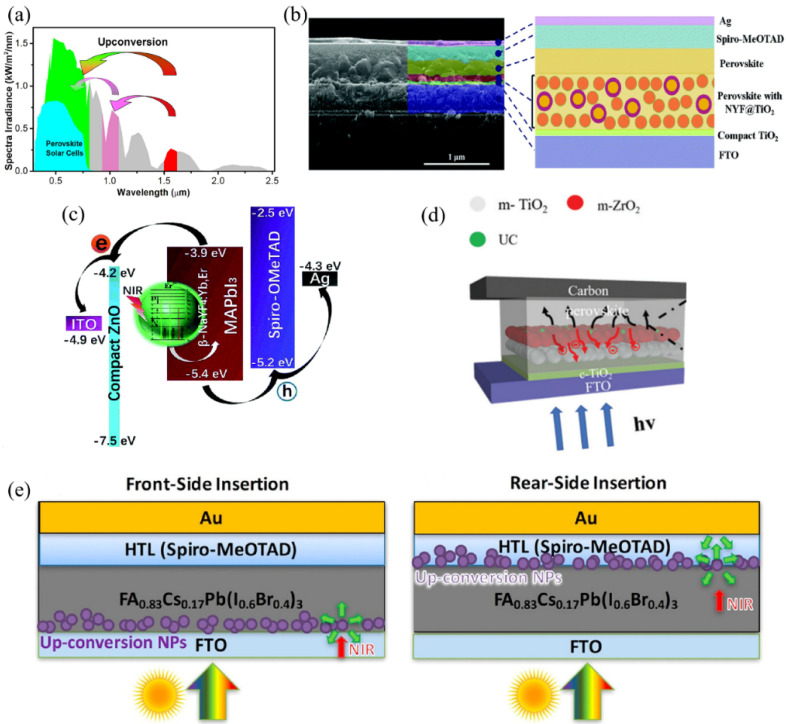


Some research groups also integrated UCNPs into a hole transfer layer (HTL). For example, NaLuF_4_:Yb,Er@NaLuF_4_ core-shell nanoparticles were prepared by using a modified two-step co-precipitation process and then incorporated into PTAA layer in the device of FTO/TiO_2_/γ-CsPbI_3_/UCNPs-doped PTAA/Au, as seen in Figure 3a. Although the absorption spectrum was broadened to NIR range, the contribution of up-conversion effect to the PCE was almost negligible which was attributed to the ultralow luminescence efficiency of UCNPs. From the reflectance and absorption spectra, the researchers found that the light scattering effect played the predominant role, which prolonged the optical path length and enhanced the photoelectric current [29]. Huang’s group introduced Li(Gd,Y)F4:Yb,Er up-conversion nanophosphors into Spiro-OMeTAD hole transport layer. The highest PCE of the FTO/cTiO_2_/mTiO_2_/CH_3_NH_3_PbI_3_/Li(Gd,Y)F_4_:Yb,Er-doped Spiro-OMeTAD/Ag-Al was 18.34% with the optimized doping amount, while the bare device was only 14.69%, as seen in Figure 3b. Through the systematical measurement and analysis, the improvement of device performance was mainly attributed to the enhanced light harvesting in the range of 400–800 nm, accelerated carrier transport, and efficient charge separation/collection caused by Li(Gd,Y)F4:Yb,Er UCNPs [30]. Subsequently, Song et al. reported a complicated core-shell structure, which combined IR-783 dye molecules, NaYF_4_:Yb^3+^,Er^3+^@NaYF_4_:Yb^3+^,Nd^3+^ UCNPs and Au nanorods (AuNRs). The energy- matched dye was used as an antenna to absorb NIR photons and the Au nanoparticle was used for local surface plasmonic resonance (Figure 3c). As a result, the up-conversion luminescent intensity was increased about 120 times. The PCE of the device (FTO/SnO_2_/AuNRs-UCNPs-IR-783 dye/perovskite/Spiro-OMeTAD/Au) boosted to 20.5% under AM 1.5G simulated sunlight irradiation, created the highest record for the up-conversion perovskite solar cells based on rare-earth metal ions up to the report date [31].

For rare-earth-based up-conversion materials used in perovskite solar cells, the hosts of Ln^3+^ ions, including single crystals of NaYF_4_ and various core-shell structures, can be placed in various parts of the solar cell, including the electron/hole transporting layers, and the perovskite layer. Different combinations have their own advantages, but the most popular one is that the core-shell structure of NaYF_4_ is integrated in the TiO_2_ layer of the device. The unique advantage is that its energy level is well-matched to the electron transport layer very well, thus reducing the obstruction to the original electronic transport capability of the device.

As it can be seen, the MYF_4_ family is a kind of well-host of Ln^3+^-doping materials. In addition, the ETL can also be the host of Ln^3+^ dopant. Each host has its own merits and peculiarities. Most reports about Ln^3+^ doping have demonstrated the MYF_4_ family as the host, which is due to that the MYF_4_ family has excellent chemical and thermal stability, thus making Ln^3+^-doped MYF_4_ show good tolerance in various preparation processes in devices. Moreover, the high transmittance and low phonon energy ensure that the introduction of Ln^3+^-doped MYF_4_ layer will not have negative effects on other layers in the devices. In perovskite solar cells, the characteristics of the ETL have a great impact on the efficiency of the devices. The various doping and modified methods have been used to improve the carrier extract and transfer efficiency of ETL. Ln^3+^ ion-doping is a good way for ETL optimization, which endows the ETL with a new ability, that is, the IR light is converted to the visible light that the perovskite absorption layer can absorb. Specific device structure, doping hosts, and performance improvements are shown in Table 1.

### 3.2. Ln^3+^-Based Down-Conversion Materials

Rare-earth metals with 4f electronic structure can also be used as down-conversion materials which have the ability to absorb ultraviolet light and re-emit visible light. In 2014, Khan and coworkers prepared YVO_4_:Eu^3+^ nano-phosphor layer on the bottom surface of perovskite solar cell (YVO_4_:Eu^3+^/Glass/FTO/cTiO_2_/mTiO_2_/CH_3_NH_3_PbI_3_/HTM/Au, Figure 4a). On the one hand, the use of down-shifting material converted a part of UV spectrum to visible region, thus, to improve the short wavelength spectral response (300–400 nm). On the other hand, the capping layer could protect device from high energy UV radiation, thus, to reduce UV-induced device degradation [43]. Similar to up-conversion materials, down-conversion materials are integrated into the electron transporting layer. Kang et al. investigated the Au@Y_2_O_3_:Eu^3+^ dual-functional films combining the effects of wavelength down-conversion and localized surface plasmon resonance for the perovskite solar cells in 2017. The photocurrent density increased from 20.7 mA/cm^2^ to 21.5 mA/cm^2^ for the Au@Y_2_O_3_:Eu^3+^-modified device (Figure 4b), which was also accompanied by improved stability [44]. Huang et al. used down conversion CeO_2_:Eu^3+^ nanophosphor to improve the device performance, and the optimal CeO_2_:Eu^3+^ was embedded into mesoporous TiO_2_ layer. The device exhibited about 6.9% PCE enhancement and significantly slower decay toward UV light irradiation when compared with bare TiO_2_-based device [45]. Another samarium (Sm)-based down-conversion material, Sr_2_CeO_4_:Sm^3+^, was introduced in the PSCs to reduce photo-loss and photo-degradation, by Chi and cooperators. The Sr_2_CeO_4_:Sm^3+^ nanophosphors could convert UV-light in the range of 283~400 nm to visible light. The optimized device of FTO/cTiO_2_/Sr_2_CeO_4_:Sm^3+^/(CsFAMA)Pb(Br,I)_3_/Spiro-OMeTAD/Au achieved the highest PCE of 17.9%, which was about 16.2% enhancement compared with the control device without optimization. Moreover, the device with Sr_2_CeO_4_:Sm^3+^ could maintain higher stability when exposed to UV-light irradiation (Figure 4c) or stored under ambient environment conditions for a much longer period [46]. Song’s group explored CsPbCl_1.5_Br_1.5_:Yb^3+^,Ce^3+^ nanocrystals as a down-conversion material which can convert blue/UV photons into lower-energy photons with PLQYs exceeding 100% for commercial silicon solar cells. Then, the nanocrystals were self-assembled in front of the device through a liquid-phase depositing method, and Figure 4d exhibits the device structure. The co-doped nanocrystals exhibited a strong 988 nm near-infrared emission from the ^2^F_5/2_–^2^F_7/2_ transition of Yb^3+^ ions, as well as the excitonic emission of CsPbCl_1.5_Br_1.5_ nanocrystals. The co-doping of Ce^3+^ can promote the process of quantum-cutting emission through energy transfer (electrons on the conduction relax to the 5d state of Ce^3+^ ions and then transfer to the Yb^3+^ ions). As a result, the highest PLQY of 146% can be achieved. The device PCE increased from 18.1% to 21.5% [47]. Later, the same group explored co-doping and tri-doping lanthanide ions in CsPbCl_x_Br_y_I_3__-x__-y_ quantum dots to improve the quantum-cutting efficiency. It was found that the Yb^3+^-Pr^3+^ and Yb^3+^-Ce^3+^ pairs can effectively sensitize the emission of Yb^3+^, which was due to the similar intermediate energy states of Pr^3+^ and Ce^3+^ with the exciton transition energy of perovskite QDs. As a result, Yb^3+^-Pr^3+^-Ce^3+^ tri-doped CsPbClBr_2_ QDs exhibited the highest PLQY of 173%, and 20% improvement in PCE can be achieved by introducing tri-doped QDs into a commercial CIGS solar cell. The PCE of devices based on different tri-doped perovskite QDs is summarized in Figure 4e [48]. Gamelin and co-workers utilized Yb^3+^:CsPb(Cl_1__-x_Br_x_)_3_ as a quantum-cutting (QC) layer in single-junction PVs, and demonstrated the potential efficiency gains through detailed balance calculations. They found that the increased PCE was mainly attributed to the reduced reflection, thermalization, and nonradiative-recombination losses from UV and blue photons. Their calculations also revealed that Yb^3+^:CsPb(Cl_1__-x_Br_x_)_3_ boosted performance in widely different geographic locations with substantially different spectral irradiances by combining PL saturation and real-world photon fluxes. The interplay between the QC PLQY, QC/PV optical coupling, and optimized QC and PV energy gaps is another key insight into the calculations, which can provide particle design rules for optimized QC/PV devices based on real-world materials and conditions [49]. Additionally, lanthanide ions were also reported to be used as interfacial modifiers in the PSCs. For example, Eu(TTA)_2_(Phen)MAA was inserted at the interface of m-TiO_2_/perovskite, and the device efficiency can be improved from 17% to 19.07% via effectively utilizing the incident UV light. The device structure and energy-level diagram are shown in Figure 4f. The inter-layer can also inhibit the device of UV-initiated degradation, thus, to effectively improve the device photostability [50].

The influence of down-conversion materials on perovskite solar cell is mainly reflected in the efficient conversion of ultraviolet light into visible light, improved light captures ability, and long-term stability. Moreover, the integration of down-conversion materials acts as light scatter during the propagation process inside the device, increasing the optical length and promoting the light absorption; these features improve the conversion efficiency of the device, as shown in Table 2.

### 3.3. Optimization of Electron Transporting Layer or Perovskite Layer by Ln^3+^ Ions-Doping

Apart from the above summarized up- and down-conversion materials, Ln^3+^ ions can also be used as dopants in perovskite materials and devices to optimize the electronic properties. One common strategy is doping Ln^3+^ ions into the charge transport layer, thus, to tune the energy-level alignment or control the interfacial defects. In 2016, in the PSC, La^3+^ ions were used to tune the band alignment of TiO_2_ layer through eliminating oxygen species and surface-inducing vacancies. The Fermi energy upward shifted from −4.55 eV to −4.43 eV for doped-La/TiO_2_, as shown in Figure 5a, thus enhancing the open circuit voltage (*V*_oc_) and fill factor (FF) [52]. Soon afterwards, a similar effect of tuning the Femi energy level can be realized by doping the Sm^3+^ rare-earth element into the TiO_2_ electron transfer layer [53], and the results of the improved device performance can be seen in Figure 5b. Wu and co-workers improved the device efficiency to a champion value of 21.75% and good UV stability by using Sc^3+^-tailored brookite TiO_2_ mesoporous layer [54]. Other rare-earth elements, such as Er^3+^ ions, were also demonstrated to be doped into the TiO_2_ layer in the CH_3_NH_3_SnI_3_-based PSCs [55].

Another important strategy is doping lanthanide ions in perovskite films or lattices. For example, Tang et al. introduced Sm^3+^ ions into CsPbBr_3_ lattice and applied it into the device architecture of FTO/cTiO_2_/mTiO_2_/CsPb_0.97_Sm_0.03_Br_3_/carbon. The strong interaction between Ln^3+^ and Br^−^ can change the surface energy during the crystal growth, thus facilitating to large formation of grains. Due to the increased grain size, prolonged carried lifetime and reduced charge carrier recombination, an ultrahigh *V*_oc_ of 1.594 V can be achieved and the highest PCE was improved from 6.99% to 10.14% [56]. A similar doping strategy was realized by Patil and co-workers, who doped Sm^3+^ ion into CsPbI_2_Br perovskite to improve the film quality. The device of FTO/cTiO_2_/mTiO_2_/CsPb_0.97_Sm_0.03_I_2_Br/P3HT/Au (Figure 5c) exhibited superior PCE and long-term stability [57]. Zhou and cooperators utilized Eu^3+^-Eu^2+^ ion pair redox shuttle to decrease I^0^ defects and oxide Pb^0^ in a cyclic transition, following the reactions of (2Eu^3+^+Pb^0^→2Eu^2+^+Pb^2+^) and (Eu^2+^+I^0^→Eu^2+^+I^0^→Eu^3+^+I^−^). As a result, the perovskite device can retain 93% efficiency under continuous 1 sun illumination or 91% PCE through thermal treating at 85 °C after 1000 h [58]. Later, Eu^3+^ ions were incorporated into inorganic CsPbI_2_Br perovskite, which can also reduce the charge recombination centers and prolong the crystal sizes as the above-mentioned Sm^3+^. Eventually, the thermal stability under 85 °C and moisture stability under the humidity of 40% can be significantly improved [59].

Recently, Chen’s group inserted an ultrathin Eu-MOF (metal-organic framework) layer as an interfacial modification layer in the device of ITO/SnO_2_/Eu-MOF/FAMACsPb(I, Br)_3_/Spiro-OMeTAD/Au. The Eu-MOF layer has multiple effects on the device properties, including improving the light absorption, changing the residual tensile strain into compressive strain and passivating the deep defect states. Through the synergistic impact induced by Eu-MOF, a highest PCE of 22.16% and long-term stability over 2000 h were achieved [60].
Figure 5(**a**) Energy-level diagram for the device of FTO/La^3+^-doped TiO_2_/mp-TiO_2_/CH_3_NH_3_PbI_3_/Spiro-OMeTAD/Au (Reprinted/adapted with permission from Ref. [52]. Copyright 2016 American Chemical Society); (**b**) *J*-*V* curves of PSCs with and without doping Sm^3+^ ions (Reprinted/adapted with permission from Ref. [53]. Copyright 2017 American Chemical Society); (**c**) Cross-section of SEM image and *J*-*V* curve for the best perovskite device of FTO/cTiO_2_/mTiO_2_/CsPb_0.97_Sm_0.03_I_2_Br/P3HT/Au (Reprinted/adapted with permission from Ref. [57]. Copyright 2020 American Chemical Society).
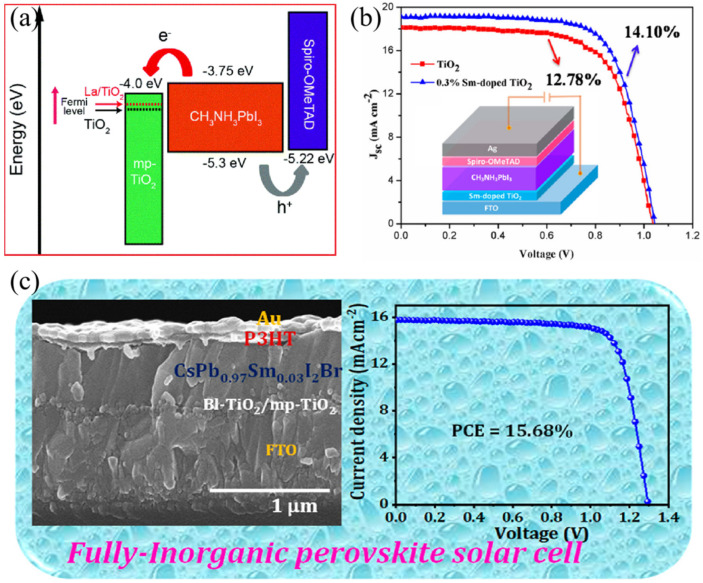


## 4. Application of Ln^3+^ in Perovskite Light-Emitting Diodes

Rare-earth doping changes the emission spectra of the perovskite materials, which can be divided into the blue shift, enhancement of the original perovskite luminescence and the new peak excited by the doped ions [61,62,63,64,65,66]. The temperature and doping concentration will affect the luminescence characteristics [67,68]. Using Ln^3+^-doping to modulate the luminescence characteristics of perovskite can expand the application of perovskite materials in light-emitting devices.

### 4.1. Adjusting the Luminescence Spectrum

It is well-known that the inferior performances of blue-violet and near-infrared emission based on perovskite materials will hinder their future commercial applications in multi-color displays. Therefore, it is important to improve the performances of the entire visible spectral region. There are many approaches to modulate the luminescence range of PVK, including doping of various ions or passivators, controlling the lattice size of PVK, or changing the composition proportion of perovskite, etc. This section focuses on the luminescence range of perovskite materials modulated by doping Ln^3+^ ions. The modulation of perovskite spectrum can be realized by doping rare-earth ions into the perovskite lattice. In order to investigate the effect of rare-earth ion-doping on the spectral position of emission and the corresponding changes in optical characteristics, Song et al. systematically demonstrated the doping of various Ln^3+^ ions (Ln^3+^ = Yb^3+^, Er^3+^, Dy^3+^, Tb^3+^, Eu^3+^, Sm^3+^, Ce^3+^) in CsPbCl_3_ nanocrystals in 2017. As it can be seen in Figure 6a, with the increase of the atomic number of the doped Ln^3+^ ions, from Ce to Yb, the primary excitation peak was gradually blue-shifted, which was due to the enlarged band gap of the perovskite host induced by the lattice contraction of doped NCs. The photoluminescence quantum yield for the Ln^3+^-doped CsPbCl_3_ NCs was enhanced, which was ascribed to the intrinsic emission of Ln^3+^ ions. In addition, Yb^3+^-doped CsPbCl_3_ NCs also exhibited strong NIR emission at around 1000 nm with high PLQY of 143% [69]. Then, they also doped different Ln elements (La, Y, Eu, Lu) into mixed K_x_Cs_1-x_PbCl_3_ quantum dots (QDs). As a result, the luminescence efficiency around 408 nm boosted to 31.2% for K_x_Cs_1-x_PbCl_3:_ Eu^3+^ QDs from 10.3% for pristine K_x_Cs_1-x_PbCl_3_ QDs [61]. Soon afterwards, Gamelin et al. specially studied the quantum-cutting effect of Yb^3+^-doped CsPbCl_3_ NCs by using variable temperature photoluminescence (PL), transient absorption, and time-resolved photoluminescence (TRPL) measurements. The doped Yb^3+^ ions can induce a shallow defect level in the perovskite lattice, which is comparable with native defects for trapping photo-excited charge carriers. In the case of Yb^3+^-doped NC (see Figure 6b), two neighboring excited Yb^3+^ ions were formed after energy transfer was captured by Yb^3+^-induced defect from the C_S_PbCl_3_ excited states. Such nonradiative energy transfer process happened at a picosecond time scale [70]. They also found that efficient quantum cutting can be realized in the Yb^3+^-doped CsPb(Cl_1-x_Br_x_)_3_ thin films, and that the PLQY exceeded 190% for the NIR emission. This work also indicated that this quantum-cutting effect is only dependent on the Yb^3+^-doped perovskite composition rather than the perovskite crystal morphologies [71]. In addition, for Pb-free metal halide double perovskite, the doping of Ln^3+^ ions also show well-modulated characteristics of emission wavelength. Yb^3+^ ions were to be incorporated into Cs_2_AgInCl_6_ double perovskite crystals (Figure 6c), resulting in 994 nm NIR emission [72]. Later, Ishii and Miyasaka used Yb^3+^-doped CsPbCl_3_ perovskite to obtain bright NIR light-emitting diode (LED, Figure 6d), and the maximum EQE around 1000 nm of the device is up to 5.9% [73]. Additionally, Yb^3+^ ions were also used to tune the fluorescent emission in lead-free double Cs_2_AgBiCl_6_ and Cs_2_AgBiBr_6_ perovskite NCs [67]. Recently, Miao and Han et al. doped Er^3+^ ions into the lead-free Cs_2_NaEr_1-x_B_x_Cl_6_ (B is In, Sb or Bi) perovskite NCs, thus, to obtain effective NIR emission at a telecommunication wavelength of 1543 nm. Such NIR emission can be mainly attributed to the energy transition from ^4^I_13/2_→^4^I_15/2_ to Er^3+^ [74]. Recently, ultrasmall CsPbX_3_ QDs formed on the surface of NaYF_4_:Yb/Tm@NaYF_4_:Yb core-shell UCNPs (see in Figure 6e) and could exhibit tunable down-conversion PL and up-conversion PL under UV and NIR excitation, respectively, which can be potentially applied in the field of fluorescent anticounterfeiting technology [75].

The influence of Ln^3+^ dopant on perovskite luminescence was mainly reflected in the change of the emission peak positions of pristine perovskite. For different kinds of rare-earth ions, the lengths of blue shift for emission peaks of pristine perovskite were various. With the increase of doping concentration, the emission peaks exhibited further blue shift. As the doped Ln^3+^ ions modified the internal defects of the original perovskite lattice, the overall performance of photoelectric and stability was enhanced.

### 4.2. White Light Emission

The Ln^3+^ ion co-doping can introduce new peaks in addition to the original perovskite excitation peak. Through the design of a particular structure, the purpose of white light emission can be achieved. White light emission and device based on rare metal ion pairs co-doped CsPbCl_3_ NCs was reported, and the corresponding ion pairs were Ce^3+^/Mn^2+^, Ce^3+^/Eu^3+^, Ce^3+^/Sm^3+^, Bi^3+^/Eu^3+^, and Bi^3+^/Sm^3+^, as illustrated in Figure 7a. Here, Ce^3+^ ions not only act as the role of blue and green emitted components, but also sensitize the red emission of Mn^2+^, Eu^3+^, and Sm^3+^ ions. The optimal white light emission with the maximum PLQY value of 75% was achieved for the sample of Ce^3+^/Mn^2+^ co-doped CsPbCl_1.8_Br_1.2_ NCs. Then the white light-emitting diode (WLED) combined co-doped perovskite NCs, GaN LED chip, and polystyrene was fabricated. The luminous efficiency and color rendering index (CRI) was 42 lm/W and over 90, respectively [76]. Considering that CsPbX_3_ (X = Br, Cl, I) have a well luminescence in green and red parts, the white light emission can be realized by introducing the blue part through Ln^3+^-doping. Monserrat and Zhang et al. prepared CsPbBr_3_ nanocrystals by introducing Nd^3+^ ions as Pb-site dopants to achieve highly efficient blue emission. They explained the enhanced mechanism via theoretical calculation. The exciton binding energy changed after partial Pb-site replacement with Nb. On the one hand, the dopants can make the perovskite lattice constrictive and Pb-Br bond shortened, thus, to enhance exciton oscillator strength. On the other hand, the flattened valence and conduction bands increased the electron and hole effective masses. Both of these caused the enhancement of PLQY. Finally, an all-inorganic perovskite WLED based on Nd^3+^-doped CsPbBr_3_ NCs, CsPbBr_3_ NCs, and CsPbBr_1.2_I_1.8_ NCs, which acted as blue-, green-, and red-emitted components, respectively, was prepared [77].

Recently, Giri and coworkers integrated Ce^3+^ and Tb^3+^ ions into 2D CH_3_NH_3_PbBr_3_ nanosheets to obtain highly blue-emitted MAPb_1-x_Ce_x_Br_3_ and MAPb_1-x_Tb_x_Cl_3x_Br_3x-3_ perovskite nanomaterials. The down-converter white LED-based on UV LED chip (Figure 7b), MAPb_0.3_Ce_0.7_Br_3_ and Rhodamine B exhibited CIE coordinates of (0.334, 0.326) [78]. In addition, Song’s group reported the first electroluminescence (EL) white LED device based on Sm^3+^-doped CsPbCl_3_ NCs. The device structure was ITO/ZnO/PEI/Sm^3+^-doped CsPbCl_3_/TCTA/MoO_3_/Au. The EL spectra were dependent on the doped Sm^3+^ ion concentration which nearly covered the entire visible spectrum region of 400~700 nm. The device structure and corresponding performances are exhibited in Figure 7c. When the Sm^3+^ ion concentration increased to 5.1 mmol%, the chromaticity coordinate (CIE) and CRI for the single-component WLED was (0.32, 0.31) and 0.93, respectively [79].

### 4.3. Doping for Luminescence Enhancement

Apart from tuning the emitted spectrum, rare metals were also reported to modulate the PL/EL efficiency and kinetics via doping them into perovskite materials. For example, Yb^3+^ ions were introduced into CsPb(Cl_1-x_Br_x_)_3_ perovskite nanocrystals. Meanwhile, the band gaps were tuned via changing the x value (x = 0~1). Through the quantum-cutting process induced by Yb^3+^ dopants, the maximum value of PLQY exceeded 200% [80]. As it can be seen in Figure 8a, the Ce^3+^ (103 pm) ion with similar ion radius to Pb^2+^ (119 pm) was introduced into the crystal structure of CsPbBr_3_ without forming additional trap states, and the PLQY was significantly improved. Ultrafast transient absorption and TRPL were performed to reveal the mechanism. Near band-edge states can be formed and induced by doping ions, thus, to accelerate the process of hot-exciton relaxation and exciton trapping to the band gap trap states. The average PL lifetimes were shortened after doping Ce^3+^. The nonradiative trapping states were replaced by newly appeared band-edge PL emissions, which were beneficial for the PL enhancement. The LED with the structure of ITO/PEDOT:PSS/poly-TPD/Ce^3+^-doped CsPbBr_3_ NCs/TPBi/LiF/Al exhibited a highest EQE value of 4.4%, which was comparable with the pristine device (1.6%) [63]. Mohammed et al. introduced YCl_3_ into CsPbCl_3_ NCs for surface defect passivation through a post- synthetic dual-surface treatment. The Pb-Cl ion pair vacancies and uncoordinated Pb atoms on the perovskite NCs’ surfaces could be counteracted by Y^3+^ and Cl^−^ ions, thus reducing the surface defects. As a result (Figure 8b), the PLQY of the post-treated NCs enhanced about 60 times compared with the pristine samples [81]. The surface passivation effect can also be realized in the case of PrCl_3_-doped CsPbBr_x_Cl_3-x_ QDs. The reduced nonradiative recombination centered on the NC surfaces mainly attributed to the enhancement of PLQY [82]. Song’s group co-doped La^3+^ and F^-^ ions into CsPbCl_3_ QDs to manipulate the optical performances through passivating the defects of Cl vacancies, resulting in improved PLQY of blue-violet emissions [68]. They also investigated the structural and optical variation tendency under high pressure with and without doping Eu^3+^ ions in the CsPbCl_3_ QDs. The energy transfer efficiency from perovskite to Eu^3+^ ions was improved when treated under relatively high pressure, thus, this improved the PL intensity [66]. Li et al. reported B-site doping in CH_3_NH_3_PbBr_3_ single crystal by using Er ion, and the photograph is shown in Figure 8c. A shallow defect level formed inside of the band gap of CH_3_NH_3_PbBr_3_ boosted the PL emission intensity. Lower trap density and higher charge carrier mobility was also achieved [65]. In addition, rare-earth metal can be used as an interfacial layer in perovskite LED. For example, Yb with low work function (2.6 eV) was introduced in the device of ITO/TFB/Quasi-2D perovskite/TPBi/Yb/Ag (as seen in Figure 7d), which efficiently promoted the electron injection via lowering the energy-level barrier. The device performances of EQE and maximum luminance were considerably superior to the device with a traditional Mg or Liq cathode interfacial layer-based device [83].

Recently, Cao and co-workers obtained functionalized CsPbBr_3_ QDs with dual-stimuli-responsive optical encoding properties by using Eu complex for surface modification. According to the calculation, the absorption energy of the modified QDs increased, leading to good stability. Furthermore, the PL of Eu^3+^: CsPbBr_3_ QDs exhibited good temperature and PH response, which provided the possibility for their application in the field of securing encrypted information [84]. Han et al. fabricated a highly efficient and stable Cs_2_NaScCl_6_ perovskite single crystal. The Sc^3+^-doped double perovskite showed a high PLQY of 29.05% for blue emission at the wavelength of 445 nm [85]. Detailed doping information and enhancement of luminescence efficiency are shown in Table 3.

## 5. Application of Ln^3+^ in Solar Concentrators

The concept of luminescent solar concentrators (LSCs) is to concentrate direct and diffuse solar radiation. Luminophores for LSCs absorb large fractions of the solar spectrum, then emit photons into a light-capture medium with high PLQY, but do not absorb their own photoluminescence.

For example, Wu’s group firstly proposed the concept quantum-cutting luminescent solar concentrator (QC-LSC) by using Yb^3+^-doped perovskite NCs, as seen in Figure 9a. The internal optical efficiency (*η*_int_) can be achieved to about 120% for Yb^3+^: CsPbCl_3_-based QC-LSC with a size of 25 cm^2^, which is much higher than the previous records realized by using Mn^2+^-doped QDs [19]. Similarly, Gamelin and coworkers demonstrated that Yb^3+^- doped CsPb(Cl_1-x_Br_x_)_3_ NC with large effective Stokes shift and high PLQY is one of the candidates for LSC luminophores (Figure 9b). In addition, the Yb^3+^: CsPb(Cl_1-x_Br_x_)_3_ NCs have an extremely low self-absorption rate [20].

## 6. Application of Ln^3+^ in Photodetectors

In 2019, Lee and co-workers synthesized lead-free CsYbI_3_ perovskite NCs with lower exciton binding energy than traditional Sn-based lead-free perovskite, which was beneficial for the application in the field of photodetectors. Due to the effective exciton dissociation and charge transport at the interface of CsYbI_3_/Graphene, the highest photoresponsivity and external quantum efficiency (EQE) of the device (Si/SiO_2_/(Au and Ti)/Graphene/CsYbI_3_ NCs, seen in Figure 10a) were 2.4 × 10^3^ A/W and 5.8 × 10^5^%, respectively [90]. Du et al. reported Yb^3+^, Er^3+^, and Bi^3+^ tri-doped lead-free Cs_2_Ag_0.6_Na_0.4_InCl_6_ perovskite single crystals and applied for X-ray detection and an anti-counterfeiting technique. The doping elements improved the luminescence efficiency induced by the Jahn–Teller effect. The limit of X-ray detection reached 8.2 nGy s^−1^ [91]. Zn^2+^, Yb^3+^, and Tm^3+^ co-doped CsPbF_3_ perovskite NCs were firstly fabricated for the application in the narrowband NIR photodetector. The device, based on Au nanorods array/Zn^2+^, Yb^3+^, Tm^3+^: CsPbF_3_ NCs, exhibited good performances (as seen in Figure 10b) with high responsivity of 106 A/W for the 980 nm spectral response [92]. In addition, Cr^3+^, Yb^3+^, e^3+^: CsPbCl_3_ perovskite QDs were coated onto a silicon photodetector for quantum cutting, thus, they realized a broad spectral response in the range of 200~1100 nm [16].

Bristowe and Cheetham et al. synthesized (CH_3_NH_3_)KGdCl_6_ and (CH_3_NH_3_)_2_KYCl_6_ rare-earth-based hybrid double perovskite, and analyzed the crystal structure [93]. Recently, Zhang and Ruan synthesized hybrid hetero-structured NCs (as illustrated in Figure 10c) including two components of CsPbBr_3_ QDs and NaYF_4_:Yb, Tm UCNPs. The prepared NCs exhibited UV-excited green emission and NIR-excited UV-blue emission [94].
Figure 10(**a**) Device structure of perovskite photodetector based on CsYbI_3_/Graphene (Reprinted/adapted with permission from Ref. [90]. Copyright 2019 John Wiley and Sons); (**b**) Device configuration and photocurrent of the photodetectors based on CsPbF_3_:Zn^2+^-Yb^3+^-Tm^3+^(or Er^3+^)/Au nanorods (Reprinted/adapted with permission from Ref. [92]. Copyright 2020 Elsevier); (**c**) Schematic illustration for the formation of heterostructures CsPbBr_3_-NaYF_4_:Yb and Tm nanocrystals (Reprinted/adapted with permission from Ref. [94]. Copyright 2021 Springer Nature).
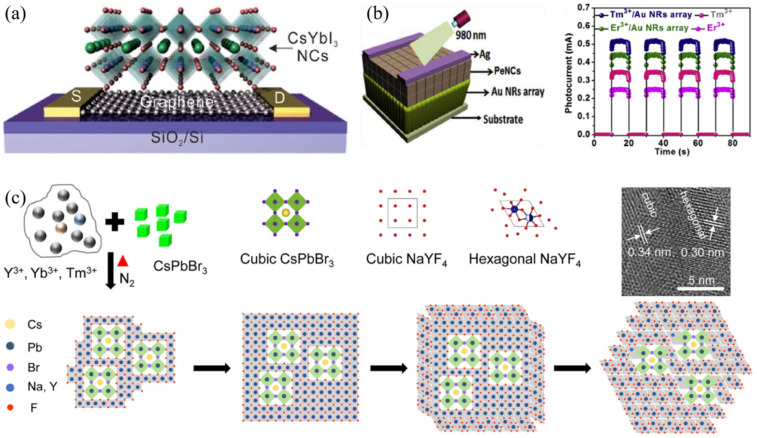


## 7. Conclusions and Outlook

In this study, we reviewed the recent development of rare-earth metal-modified metal halide perovskite materials and their corresponding optoelectronic devices. We discovered that different Ln^3+^ ions can absorb and emit light both in the UV and NIR regions, thus making up the deficiency of utilization for solar spectrum by perovskite materials. Therefore, Yb^3+^, Er^3+^, Tm^3+^, Sc^3+^, Nd^3+^, and Ho^3+^ ions-based up-conversion nanoparticles were incorporated inside or proximal to the perovskite layer in the PSCs, resulting in higher device efficiency through broadening absorption spectral range to NIR region. Similarly, down-conversion nanocrystals-doping or co-doping with Eu^3+^, Sm^3+^, Yb^3+^, Ce^3+^, and Pr^3+^ ions can convert UV-light into visible light, which simultaneously improved PCE and prevented degradation triggered by UV-light in the perovskite solar cells. Moreover, due to their unique electronic structure, Ln^3+^ ions (La^3+^, Sc^3+^, Sm^3+^, Eu^3+^, and so on) were also used as dopants to facilitate perovskite film growth, tailor the energy band alignment, and passivate the defect states, thus, to boost the device performances. In the field of perovskite light emission, Ln^3+^ ions including Yb^3+^, Er^3+^, Dy^3+^, Tb^3+^, Eu^3+^, Sm^3+^, Ce^3+^, and Tm^3+^ ions were doped or co-doped into perovskite nanocrystals to realize multi-color emissions covering the entire visible spectral region. Based on this, Ln^3+^-doped perovskite NCs were applicable for fabrication of white light-emitting diodes. Furthermore, Yb^3+^, Ce^3+^, Y^3+^, Pr^3+^, and Eu^3+^ ions were doped into perovskite lattice, which can form near band gap states, reduce Pb or halide vacancies or passivate surface defect, resulting in decreased nonradiative recombination centers and enhanced PLQY. Some Ln^3+^ ions, such as Y^3+^ ions, can also be used as an interfacial layer in the perovskite LED, which can promote the electron injection via lowering the energy-level barrier. Apart from the above two main application fields, Ln^3+^ ions have also been reported to be used in the fields of photodetectors and luminescent solar concentrators. These indicate the huge potential of rare-earth metals in improving the performances of the perovskite optoelectronic devices.

The host type of rare-earth ions determines the role of rare-earth ions in perovskite photoelectric devices. With MYF4 (M = Li, Na, K, Ru. and Cs) as host, Ln^3+^ ions-doping brings the ability of up-conversion, which enables solar cells to obtain a better life and efficiency. When the perovskite layer is used as the host, the introduction of Ln^3+^ ions reduce defects, increases efficiency, and introduces a new luminescence peak. Therefore, different hosts have completely different functions, and the future research on Ln^3+^ ions will also have new breakthroughs in the host types.

Based on these abundant research results, we suggest that additional investigations should be supplemented. (1) Although Ln^3+^ ions have been demonstrated to be effective in the perovskite light emissions, most work concentrated on the photoluminescence spectra and PLQY enhancement of perovskite materials. Extended work should focus on the device performances’ enhancement of multi-color LED by doping Ln^3+^ ions, especially NIR and UV perovskite LEDs which needs to be implemented. (2) Luminescent solar concentrators based on perovskite materials and rare-earth metals need to be systematically studied, and their applications in photovoltaic field should be explored. (3) Perovskite photodetectors with high photocurrent and narrow spectrum responsivity should be further researched with the introduction of Ln^3+^ ions.

## Figures and Tables

**Figure 1 nanomaterials-12-01773-f001:**
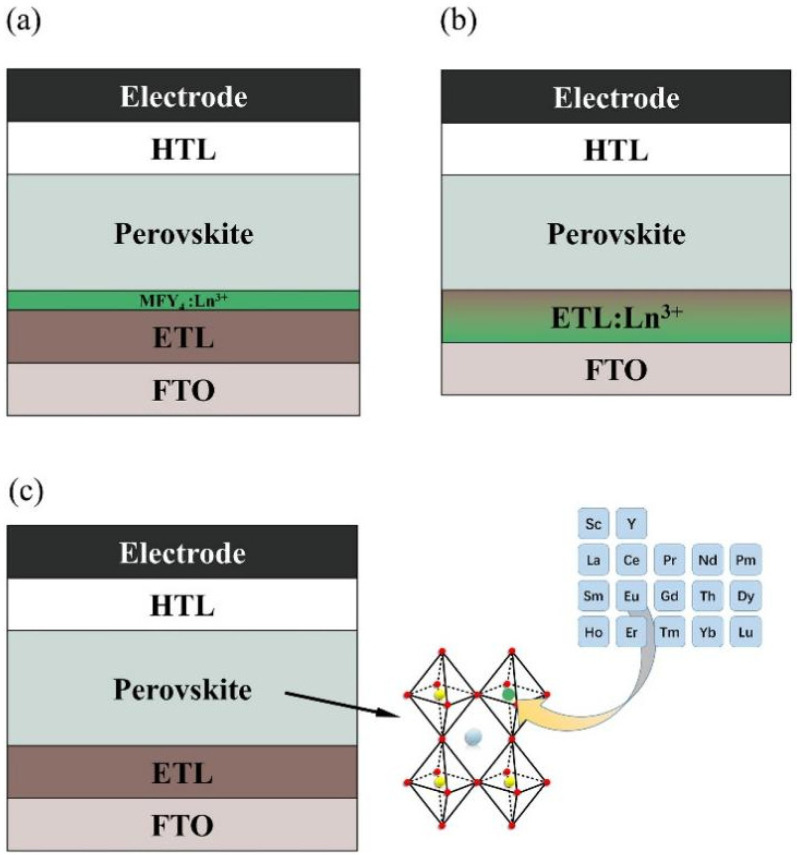
Three modified strategies of Ln^3+^ in perovskite device: (**a**) Inserting Ln^3+^-based conversion layer in perovskite device; (**b**) Ln^3+^ ion mixed into carrier transport layer; (**c**) Ln^3+^ ion doped in perovskite active layer.

**Figure 3 nanomaterials-12-01773-f003:**
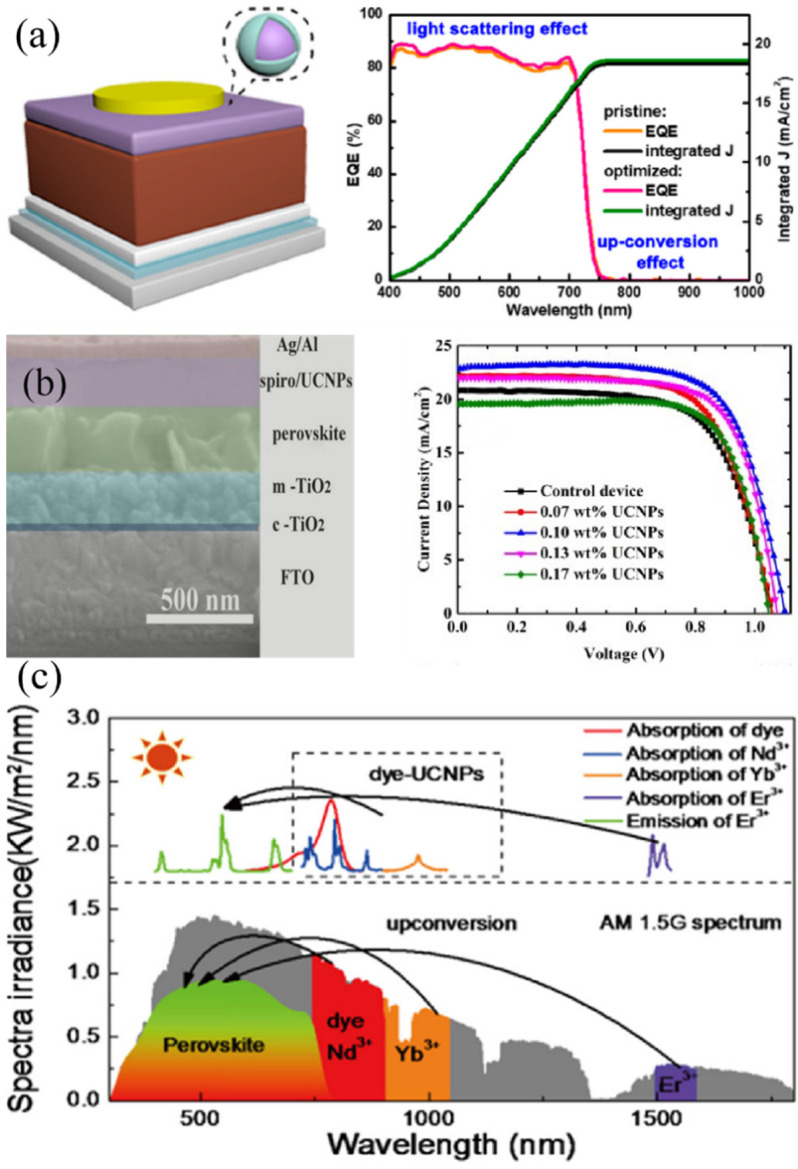
(**a**) Schematic device structure and external quantum efficiency curves of FTO/TiO_2_/γ-CsPbI_3_/NaLuF_4_:Yb,Er@NaLuF_4_ UCNPs-doped PTAA/Au (Reprinted/adapted with permission from Ref. [29]. Copyright 2019 American Chemical Society); (**b**) Cross-section SEM image for the device of FTO/cTiO_2_/mTiO_2_/CH_3_NH_3_PbI_3_/Li(Gd,Y)F_4_:Yb,Er-doped Spiro-OMeTAD/Ag-Al and *J**-**V* curves of devices with different amounts of UCNPs (Reprinted/adapted with permission from Ref. [30]. Copyright 2019 Elsevier); (**c**) The spectrum absorption range of PSCs and up-conversion spectral regions of dye-NaYF_4_:Yb^3+^,Er^3+^@NaYF_4_:Yb^3+^,Nd^3+^ UCNPs (Reprinted/adapted with permission from Ref. [31]. Copyright 2020 American Chemical Society).

**Figure 4 nanomaterials-12-01773-f004:**
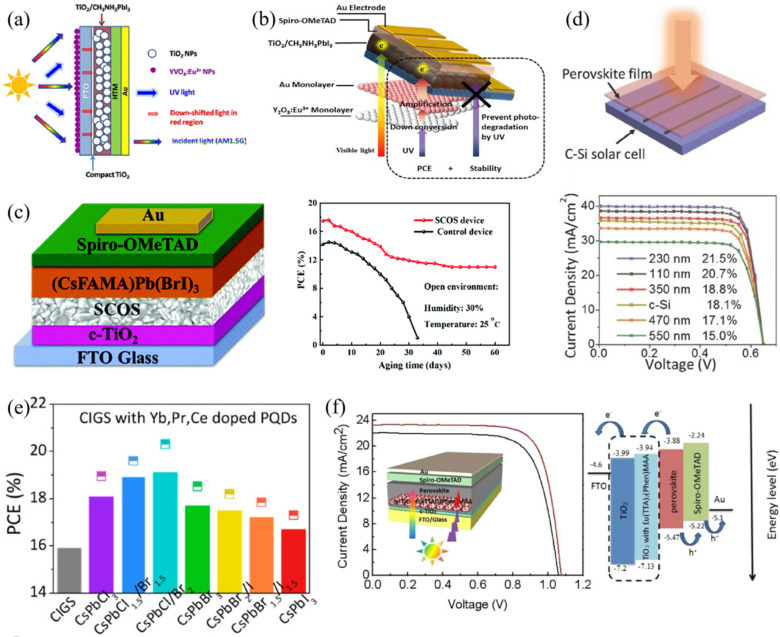
Schematic diagrams for (**a**) perovskite solar cell of YVO_4_:Eu^3+^/Glass/FTO/cTiO_2_/mTiO_2_/CH_3_NH_3_PbI_3_/HTM/Au (Reprinted/adapted with permission from Ref. [43]. Copyright 2014 AIP Publishing) and (**b**) Au@Y_2_O_3_:Eu^3+^-modified PSC with the functions of down-conversion and device stability enhancement (Reprinted/adapted with permission from Ref. [44]. Copyright 2017 Springer Nature); (**c**) Device structure of FTO/cTiO_2_/Sr_2_CeO_4_:Sm^3+^/(CsFAMA)Pb(Br,I)_3_/Spiro-OMeTAD/Au and long-term stability measurements under open environment (Reprinted/adapted with permission from Ref. [46]. Copyright 2019 Royal Society of Chemistry); (**d**) Device structure and *J*-*V* curves for silicon solar cells by using CsPbCl_1.5_Br_1.5_:Yb^3+^,Ce^3+^ as down-conversion material (Reprinted/adapted with permission from Ref. [47]. Copyright 2017 John Wiley and Sons); (**e**) PCE comparisons for CIGS solar cells with Yb^3+^-Pr^3+^-Ce^3+^ tri-doped CsPbCl_x_Br_y_Cl_3__-__x__-__y_ (x ≥ 0, y ≤ 3, x + y ≤ 3) quantum dots (Reprinted/adapted with permission from Ref. [48]. Copyright 2019 American Chemical Society); (**f**) Device structure and performance of Eu(TTA)_2_(Phen)MAA-doped PSC and the corresponding energy-level alignment (Reprinted/adapted with permission from Ref. [50]. Copyright 2019 American Chemical Society).

**Figure 6 nanomaterials-12-01773-f006:**
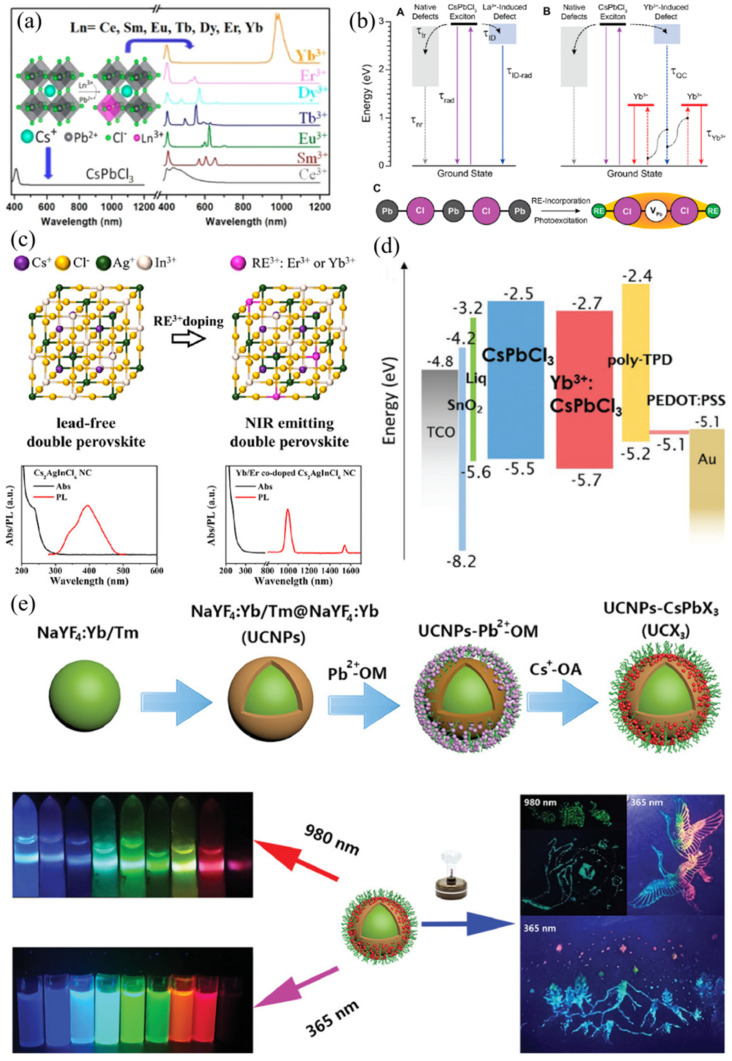
(**a**) Photoluminescence spectra of CsPbCl_3_ nanocrystals with doping Yb^3+^, Er^3+^, Dy^3+^, Tb^3+^, Eu^3+^, Sm^3+^, Ce^3+^ ions, respectively (Reprinted/adapted with permission from Ref. [69]. Copyright 2017 American Chemical Society); (**b**) Schematic maps for La^3+^- induced defect emission, Yb^3+^ sensitization mechanism, and charge-neutral vacancy-defect structure of Ln^3+^-doped CsPbCl_3_ nanocrystals (Reprinted/adapted with permission from Ref. [70]. Copyright 2018 American Chemical Society); (**c**) Crystal lattice structure of Cs_2_AgInCl_6_ perovskite with doping Ln^3+^ ions and the corresponding absorption/photoluminescence spectra (Reprinted/adapted with permission from Ref. [72]. Copyright 2019 American Chemical Society); (**d**) Energy-level diagram of perovskite LED with structure of TCO/SnO_2_/Liq/Yb^3+^:CsPbCl_3_/poly-TPD/PEDOT:PSS/Au (Reprinted/adapted with permission from Ref. [73]. Copyright 2020 Wiley-VCH); (**e**) Fabrication process of NaYF_4_:Yb/Tm@NaYF_4_:Yb UCNPs-CsPbX_3_ (X = Cl, Br, I or mixed halide elements) for full color anticounterfeiting (Reprinted/adapted with permission from Ref. [75]. Copyright 2021 John Wiley and Sons).

**Figure 7 nanomaterials-12-01773-f007:**
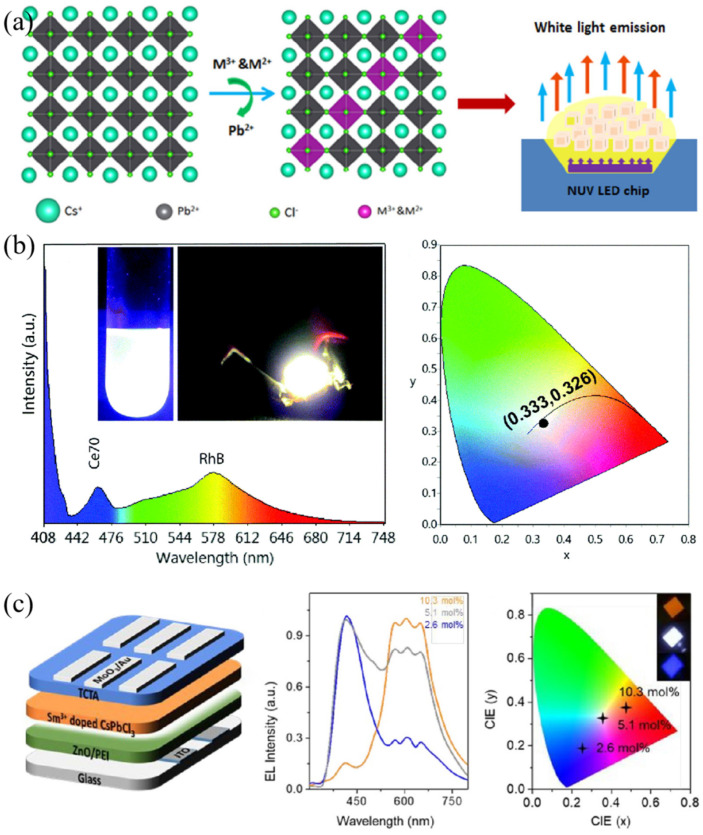
(**a**) Crystal structure of CsPbCl_x_Br_3-x_ with co-doping metal ion pairs (M^3+^/M^2+^) and the white LED combining CsPbCl_1.8_Br_1.2_:Ce^3+^/Mn^2+^ and ultraviolet chip (Reprinted/adapted with permission from Ref. [76]. Copyright 2018 American Chemical Society); (**b**) EL spectrum, white light emission photograph and CIE chromaticity coordinates of WLED based on UV LED chip, Ce^3+^:CH_3_NH_3_PbBr_3_, and Rhodamine B (Reprinted/adapted with permission from Ref. [78]. Copyright 2021 Royal Society of Chemistry); (**c**) Device structure, EL spectra, and CIE coordinates for perovskite WLED based on Glass/PEI/Sm^3+^:CsPbCl_3_/TCTA/MoO_3_/Au (Reprinted/adapted with permission from Ref. [79]. Copyright 2020 American Chemical Society).

**Figure 8 nanomaterials-12-01773-f008:**
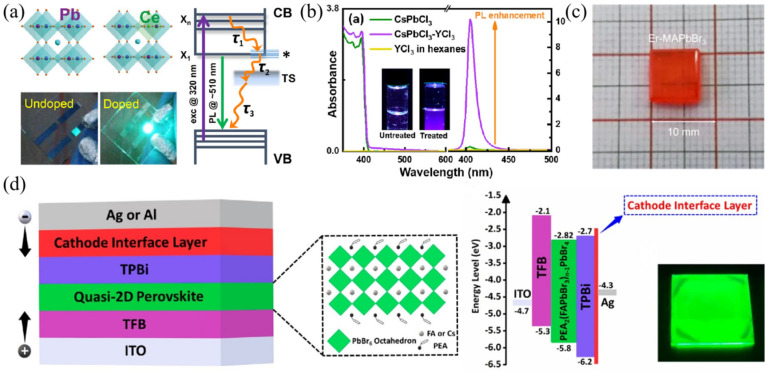
(**a**) Crystal lattice structure of Ce^3+^-doped CsPbBr_3_, LED photographs, and schematic diagram of photophysical mechanism (Reprinted/adapted with permission from Ref. [63]. Copyright 2018 American Chemical Society); (**b**) Absorption and PL spectra of Y^3+^-doped CsPbCl_3_ nanocrystals (Reprinted/adapted with permission from Ref. [81]. Copyright 2018 American Chemical Society); (**c**) Photograph of Er-doped CH_3_NH_3_PbBr_3_ perovskite single crystal (Reprinted/adapted with permission from Ref. [65]. Copyright 2020 American Chemical Society); (**d**) Yb-modified device of ITO/TFB/Quasi-2D perovskite/TPBi/Yb/Ag: device schematic illustration, molecular structure of PEA_2_(FAPbBr_3_)_n-1_PbBr_4_, device energy band alignment, and fluorescence photograph of perovskite film (Reprinted/adapted with permission from Ref. [83]. Copyright 2020 American Chemical Society).

**Figure 9 nanomaterials-12-01773-f009:**
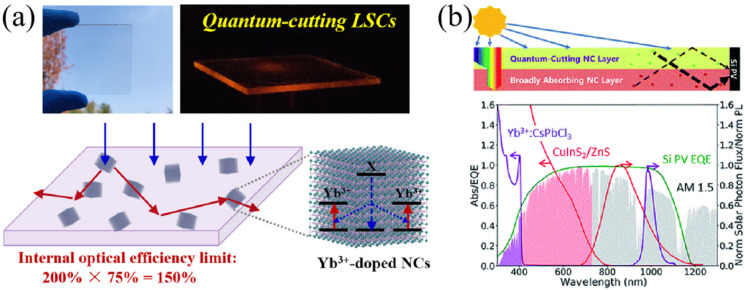
(**a**) The picture of Yb^3+^-doped CSPbCl_3_ LSC under sunlight and illustration of internal optical efficiency (Reprinted/adapted with permission from Ref. [19]. Copyright 2019 American Chemical Society); (**b**) Schematic maps of monolithic bilayer LSC and absorption/PL spectra of Yb^3+^:CsPbI_3_ nanocrystals (Reprinted/adapted with permission from Ref. [20]. Copyright 2018 Royal Society of Chemistry).

**Table 1 nanomaterials-12-01773-t001:** The PCE enhancement of up-conversion Ln^3+^-doping solar cell device.

Device Structure	Doped Materials	PCE Enhancement	Ref.
UC crystals/FTO/TiO_2_/MAPbI_3_/HTM/Au	LiYF_4_:Yb^3+^, Er^3+^	7.9% to 8.8%	[22]
FTO/compact-TiO_2_/NaYF_4_ core-shell nanoparticles-TiO_2_/MAPbI_3_/ Spiro-MeOTAD/Ag	NaYF4:Yb^3+^, Tm^3+^	14.1% to 16.9	[24]
FTO/compact-TiO_2_/NaYF_4_-TiO_2_/MAPbI_3_/Spiro-MeOTAD/Au	NaYF4:Yb^3+^, Er^3+^	13.7% to 16.0%	[17]
ITO/compact-TiO_2_/ NaYF_4_-MAPbI_3_/ Spiro-MeOTAD/Ag	NaYF4:Yb^3+^, Er^3+^	13.46% to 19.70%	[25]
FTO/compact-TiO_2_/TiO_2_-Cu_2-x_S@SiO_2_@Er_2_O_3_ /MAPbI_3_/Spiro-MeOTAD/Au	mCu_2-x_S@SiO_2_@Er_2_O_3_	16.2% to 17.8%	[26]
FTO/compact-TiO_2_/mp-TiO_2_/mp-ZrO2- NaYbF_4_ /FA_0.4_MA_0.6_PbI_3_/C	NaYF_4_:Ho^3+^	10.9% to 14.3%	[27]
FTO/KY_7_F_22_-Yb^3+^Er^3+^/FA_0.83_Cs_0.17_Pb(I_0.6_Br_0.4_)_3_/Spiro/Au FTO/FA_0.83_Cs_0.17_Pb(I_0.6_Br_0.4_)_3_/KY_7_F_22_-Yb^3+^Er^3+^/Spiro/Au	KY_7_F_22_:Yb^3+^, Er^3+^	13.5% to 14.0%	[28]
FTO/TiO_2_/CsPbI_3_/UCNPs-PTAA/Au	NaLuF_4_:Yb,Er@NaLuF_4_	15.5% to 15.9%	[29]
FTO/c-TiO_2_/m-TiO_2_/CH_3_NH_3_PbI_3_/Spiro-UCNP/Ag-Al	Li(Gd, Y)F_4_:Yb^3+^, Er^3+^	14.7% to 18.3%	[30]
FTO/SnO_2_/UCNPs-Dye-AuNRs/FAMACsPb(I, Br)_3_	NaYF_4_:Yb^3+^, Er^3+^@NaYF_4_:Yb^3+^, Nd^3+^ core-shell	19.4% to 20.5%	[31]
FTO/compact-TiO_2_/NaYF_4_/MAPbI_3_/Spiro-MeOTAD/Ag	NaYF_4_:Yb^3+^, Er^3+^	17.8% to 18.1%	[32]
FTO/compact-TiO_2_/TiO_2_ nanorods/MAPbI_3_/Spiro-MeOTAD/Au	TiO_2_:Yb^3+^, Er^3+^	10.6% to 12.9%	[33]
FTO/compact-TiO_2_/TiO_2_-ZrO_2_-NaYF_4_@SiO_2_ /MAPbI_3_/Carbon	NaYF_4_:Yb^3+^, Er^3+^	8.2% to 9.9%	[34]
FTO/compact-TiO_2_/MAPbI_3_- TiO_2_-NaYF_4_@TiO_2_ /Spiro-MeOTAD/Au	NaYF4:Yb^3+^, Tm^3+^	13.4% to 16.3%	[35]
ITO/SnO_2_/FAPbI_3_/UCNPs/Spiro-UCNPs/Au	NaYF_4_:Yb^3+^, Er^3+^	16.0% to 18.0%	[36]
FTO/TiO2/UCNPs/FACsPb(I,Br)_3_/Spiro/Au	β-NaYF_4_: Nd^3+^, Yb^3+^, Er^3+^	18.0% to 19.2%	[37]
ITO/ZnO/IR806-UCNPs-MAPbI_3_/Spiro/Ag	*β*-NaYF_4_:Yb^3^^+^, Er^3^^+^	13.5% to 17.5%	[38]
FTO/TiO_2_/UCNPs-(FA_0.83_MA_0.17_)_0.95_Cs_0.05_Pb(I_0.9_Br_0.1_)_3_/Spiro/Au	NaYF_4_:Yb^3^^+^/Er^3^^+^/Sc^3+^@NaYF_4_ core-shell	17.44% to 20.2%	[39]
FTO/Er-TiO_2_/CH_3_NH_3_PbI_3-x_Cl_x_/Spiro/Ag	Er-doped TiO_2_	9.1% to 10.6%	[40]
FTO/TiO_2_ nanorod/UCNP/MAPbI_3_/C	SiO_2_/NaYF_4_:Yb, Er@SiO_2_	11.9% to 14%	[41]
FTO/UCNP-TiO_2_/FAMACsPb(I, Br)_3_/Spiro/Au	Er^3+^-Yb^3+^-Li^+^ tri-doped TiO_2_	14.0% to 16.5%	[42]

**Table 2 nanomaterials-12-01773-t002:** The PCE enhancement of down-conversion Ln^3+^-doping solar cell device.

Device Structure	Doped Materials	PCE Enhancement	Ref.
DCNP/FTO/TiO_2_/CH_3_NH_3_PbI_3_/HTM/Au	YVO_4_:Eu^3+^	7.4% to 7.9%	[43]
FTO/cTiO_2_/mTiO_2_-CeO_2_:Eu^3+^/MAPbI_3_/Spiro/Au	CeO_2_:Eu^3+^	10.1% to 10.8%	[45]
FTO/cTiO_2_/DCNP/(CsFAMA)Pb(Br,I)_3_/Spiro-OMeTAD/Au	Sr_2_CeO_4_:Sm^3+^	15.4% to 17.9%	[46]
C-Si solar cell/Perovskite film as DCNP	Yb^3+^, Ce^3+^ co-doped Cs PbCl_1.5_Br_1.5_	18.1% to 21.5%	[47]
Down-converter for CuIn_1-x_Ga_x_Se_2_ (CIGS) and Si solar cell	Yb^3+^, Ln^3+^(Nd, Dy, Tb, Pr, Ce) doped-QDs	~20% enhancement	[48]
Down-conversion film for single-junction PV	Yb^3+^:CsPb(Cl_1-x_Br_x_)_3_	-	[49]
FTO/cTiO_2_/mTiO_2_-DCNP/(FA_0.83_MA_0.17_)_0.95_Cs_0.05_Pb(I_0.83_Br_0.17_)_3_/Spiro/Au	Eu(TTA)_2_(Phen)MAA	17.0% to 19.0%	[50]
NaYF_4_:Eu^3+^/FTO/TiO_2_/Cs_0.05_(MA_0.17_FA_0.83_)_0.95_Pb(I_0.83_Br_0.17_)_3_/Spiro/Au	NaYF_4_:Eu^3+^	17.6% to 20.1%	[51]

**Table 3 nanomaterials-12-01773-t003:** The PLQY enhancement of Ln^3+^-doping PVK and the EQE enhancement of Ln^3+^-doping perovskite LED.

Modified Materials	Main Emission Wavelength	PLQY/EQE	Ref
K^+^ and lanthanide elements-doped CsPbCl_3_ QDs	408 to 495 nm	90%	[61]
Eu^3+^ and Tb^3+^-doping CsPbBr_3_	592, 612/543, and 582 nm	-	[62]
Ce^3+^-doped CsPbBr_3_	516 to 510 nm	89%/4.4%	[63]
Er-doping MAPbBr_3_ single crystals	546 nm	-	[65]
Eu^3+^-doped CsPbCl_3_	570 to 710 nm	-	[66]
Yb3+-doped Cs2AgBiX6	900–1200 nm	0.3 %/-	[67]
CsPbCl3QDs by co-doping La3+ and F- ions	410 nm	36.5%	[68]
Various lanthanide ions	270 to 420 and 1000 nm	143%/-	[69]
Yb^3+^ CsPbCl_3_	990 nm	Over 100%	[70]
(Yb3+) CsPb(Cl_1-x_Br_x_)_3_	990 nm	190%	[71]
Yb^3+^ and Yb^3+^/Er^3+^ co-doped CsPbCl_3_	986 nm	127.8%/-	[72]
Yb^3+^-doped CsPbCl_3_	1000 nm	Over 60%/5.9%	[73]
Cs2NaEr1-xBxCl6 (B: In, Sb, or Bi; x = 0, 0.13, and 0.5)	1543 nm	-	[74]
Yb^3+^/Er^3+^ co-doped CsPbCl3 N	986 nm and 1533 nm	127.8%/-	[75]
Cr^3+^ Ce^3+^ Yb^3+^ tri-doped CsPbCl_3_ QDs	405 to 394 and 980 nm	188%/-	[16]
CsPbCl_3_: Mn^2+^, Er^3+^	580 to 600 nm	Enhanced by 100 times than origin/-	[18]
Ce^3+^ and Mn^2+^ co-doped CsPbCl_1.8_Br_1.2_	429, 460, and 592 nm	75%	[76]
Nd^3+^-doped CsPbBr_3_	459 nm	90%	[77]
Ce^3+^ and Tb^3+^-doped MAPbBr_3_	454 to 518 nm	100%/-	[78]
MAPbBr_3-x_I_x_-coated YAG:Ce^3+^ phosphors	490, 535, 550, 680, 715, 765 nm	-	[21]
Sm^3+^-doped CsPbCl_3_	565, 602, 645, and 710 nm	85%/1.2%	[79]
Yb^3+^-doped CsPb(Cl_1-x_Br_x_)_3_	979 nm	Approaching 200%/-	[80]
YCl_3_-treated	394 and 404 nm	60%/-	[81]
Pre-optimize CsPbCl_3_ using PrCl_3_	407 and 404 nm	89%/-	[82]
Yb layer	536 nm	-/16.4%	[83]
Europium complex to CsPbBr3 QDs	500, 592, and 616 nm	-	[84]
Sc-based double perovskite Cs2NaScCl6	459 and 635 nm	29.05%/-	[85]
Yb/Er-doped Cs_2_AgInCl_6_	395/996 and 1573 nm	3.6 ± 0.4%/-	[86]
Yb^3+^-doped CsPbCl_3_	980 nm	87.9%/-	[87]
Yb-doped Cs_2_AgInCl_6_	996 nm	3–4%/-	[88]
CsPbX_3_ Zeolite-Y composite	400 to 600 nm	-	[89]

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
