# Peer review of "Review for Rare-Earth-Modified Perovskite Materials and Optoelectronic Applications"

_nanomaterials, 2022, doi:10.3390/nano12101773_

Round 1
Reviewer 1 Report
The work is a review of the applications of lanthanides in perovskites to improve some of the characteristics of these materials and their applications in photovoltaics and optoelectronics. The work does not have a research character of its own, and it focuses on a rather a bit superficial overview of various earlier results. It is written quite neatly and is provided with illustrations from other works, which, however, significantly reduces their originality. The value is to put the various pieces of information together. Here, however, it should be emphasized that the use of lanthanides is not a universal paneceum for the limitations of perovskite devices - which was too optimistically presented in the review. Besides, there are many other methods of improving perovskite devices, and they should be at least mentioned (with appropriate references), if not in a separate paragraph then at least in the introduction. It should also be noted that the use of lanthanides does not lead to record improvements compared to other methods (as shown in the examples presented in the review).
Therefore, the work requires some supplementation to show the role and effectiveness of the use of lanthanides in comparison to other methods. Important is also to show how lathanide admixteres degrade some electronic properties (like mobility of carriers or recombination rate). The use of plasmonic nano-converters is more effective in the field of perovskite solar cells and their accomodation to sun-light spectrum. The increase in efficiency is even up to 40% of the relative efficiency due to nano metallization of perovskite multilayer,
https://doi.org/10.1016/j.nanoen.2020.104751
and references ibid.
Similarly, changes in the band gap width, which is important from the point of view of the constraints imposed by the Shockley–Queisser limit, can be achieved in wide range by varying the chemical composition (and not by lanthanide admixtures),
https://doi.org/10.3390/ma1506225
and references ibid.
It seems advisable to include at least an extensive paragraph with comparisons to other methods of improve perovskite to actually emphasize the relative importance of the proposed method of lanthanide admixtures .
Reviewer 2 Report
Li et al. has summarized a review report as “Review for Rare-Earth Modified Perovskite Materials and Optoelectronic Applications.” Though throughout the review, I find only the summary of different optoelectronic applications without much detail on the material physics/chemistry. Overall, I find the review a concise report on different optoelectronic applications of rare earth metal ion/material incorporated metal halide perovskites. However, the writing and arrangement of various sections need substantial improvement. After all the necessary changes, the manuscript can be further considered for publication. I have attached my comments/suggestions below.
- The authors have reported two kinds of integration of rare earth metals into metal halide perovskites mainly as a separate filtering/sensitizing layer in solar cells and doped ions into an active or transport layer. In the introduction, the authors have stated “Rare earth (RE) metals with triply oxidized state (lanthanide ions, Ln3+) possess different kinds of energy transitions, which determining that they can emit fluorescence in a wide wavelength range covering from ultraviolet to intermediate infrared regions [8].” Reference 8 itself is an elaborate review on the doping of rare earth metals and their physical properties. It could have been much better if the authors started the review with the necessity of rare-earth doping with some schematic diagrams. These need not always be adapted from previous reports! Then the corresponding physical mechanisms should have been explained for the reported two cases, (1) layer-based sensitization vs. (2) doped sensitization.
- The authors started the review report on the application of rare earth material in perovskite solar cells in section 2. The section starts like “As is well-known, due to the intrinsic band gap of perovskite materials, perovskite 70 based solar cells are usually unable to utilize light beyond the visible region (the range of UV and NIR light), thus limiting further development of the device efficiency”. It is not only the case of perovskite but single-junction semiconductor solar cells are unable to absorb infrared photons. Probably authors should write a few sentences on the Shockley-Queisser limit of single-junction solar cells and why it is important to utilize the infrared photons. Then they should come to the importance of lanthanides!
- In this section, the authors reported two sub-sections on upconversion systems and down conversion systems, respectively. I feel totally lost in so many who did what reports! I strongly feel the necessity of summarizing these into proper table format with device architectures and the corresponding efficiency improvement! The use of the MYF4 lanthanide family as an up or a down-converted sensitizing layer in solar cells is attractive. The author may highlight some properties of these lanthanides first and then highlight the reported literature. Could the authors highlight some of the important aspects when MYF4 is dispersed in the electron transport layer vs utilized as a separate layer?
- Figure 1 is not aligned with its caption and the text.
- The second subsection of section 2 is on down conversion material. Here, also a proper tabulation of reported down converters in various solar cells is required.
- Figure 3 is not aligned with the caption and the text.
- In subsection 3 of section 2 though the title reads as “Optimization of electron transporting layer with Ln3+ ions”, the authors describe doping of Ln3+ ions not only in electron transport layer TiO2 but also in active perovskite absorber.
- Section 3 is on the Ln3+ effect on light-emitting diodes with three subsections. In the first subsection “Adjusting the luminescence spectrum”, the authors mentioned “It is well-known that the inferior performances of blue-violet and near-infrared emission based on perovskite materials will hinder their future commercial applications in multi-colour displays. ….. The modulation of perovskite spectrum can be realized by doping rare earth ions into the perovskite lattice.” However, when the PLQY enhancement of the host lattice could be achieved even with doping earth-abundant materials then why should one focus on ‘expensive rare earth materials? The author should rephrase to emphasize the actual outcomes.
- In subsection 2. “White-light emission”, the authors stated, “The Ln3+ ion co-doping can introduce two more peaks in addition to the original perovskite excitation peak to achieve the purpose of white light emission.” This is not always the case! The appearance of two additional peaks is not always happening, maybe only in certain cases!
- Line 431, section 3.3 , “Untrafast transient absorption…..” should be ultrafast
- Figure 7 is not aligned with the text and the captions
- In the LED section, please provide a table with the device architecture and the EQE and/ PLQY numbers after Ln3+ induced modifications
- Figure 8 is not aligned with the text and the captions
- Should not the structure analysis of the Ln3+ doped perovskite be at the beginning of the review rather than in the last section?
Round 2
Reviewer 1 Report
the submission has been improved
Reviewer 2 Report
The authors have now significantly improved the manuscript following the suggestion given in the earlier review. I recommend it for publication in nanomaterials.